# Caregiver-reported social impacts in down syndrome regression disorder

Katherine N. Chow[1], Lilia Kazerooni [1], Maeve C. Lucas[1], Samuel T. Otey [1], Mariam M. Yousuf[1], Ruth Brown[2], Eileen A. Quinn[3], Jonathan D. Santoro [1,4]*

**1** Division of Neurology, Department of Pediatrics, Children's Hospital Los Angeles, Los Angeles, California, United States of America, **2** Virginia Institute for Psychiatric and Behavioral Genetics at Virginia Commonwealth University, Richmond, Virginia, United States of America, **3** Division of Developmental and Behavioral Pediatrics, University of Toledo College of Medicine and Life Sciences, Toledo, Ohio, United States of America, **4** Department of Neurology, Keck School of Medicine of the University of Southern California, Los Angeles, California, United States of America

* jdsantoro@chla.usc.edu

## Abstract

### Background

Down Syndrome Regression Disorder (DSRD) is an acute neurocognitive regression in individuals with Down syndrome (DS), causing a profound loss of acquired skills. DSRD increases demands on caregivers, to sleep disturbances, financial distress, and negative impacts on caregiver-reported social connections and perceived social support. The goal of this study was to characterize the caregiver-reported impacts of DSRD on social relationships by comparing their experiences to those of caregivers of individuals with DS and other neurological disorders (DSN).

### Design/methods

This is a narrative burden-of-care study, not a network study. Using cross-sectional study design, caregivers of individuals with DSRD (n = 228) and DSN (n = 137) were recruited from a neurology clinic and a DSRD Facebook support group. Participants completed the DSRD Caregiver Distress Survey (CDS), which included four qualitative, open-ended questions focused on self-perception of adult friendships, social relationship impact, spouse/partner impact, and perceived shrinkage of social world. Responses were analyzed using thematic coding; resulting theme frequencies summarize caregiver-reported perceptions and narratives and do not represent objectively measured social network structure.

### Results

In the DSRD cohort, a high-level overview revealed that 65.66% of responses reported a negative impact on adult friendships, while 71.21% reported a negative impact on social relationships. A negative impact on spouse/partner relationships was reported in

**Data availability statement:** All data files are available from the Dryad database (accession number(s) WTKFXToG-x5dKYi__YrQhbCQOPb5ktLICY_hd5mIBBU). Please reference the link below. http://datadryad.org/share/LINK_NOT_FOR_PUBLICATION/WTKFXToG-x5dKYi__YrQhbCQOPb5ktLICY_hd5mIBBU.

**Funding:** The author(s) received no specific funding for this work.

**Competing interests:** The authors have declared that no competing interests exist.

51.53% of responses, and a perceived shrinkage of social world was found in 52.82%. Caregivers in the DSRD group were significantly more likely to report "Social Withdrawal and Isolation" (43.2% vs. 17.9%, p = 0.006), "Loss of Community Participation and/or Support" (16.7% vs 4.5%, p = 0.043) and a "Perceived Enduring Loss of Social Connections" (35.3% vs. 8.7%, p = 0.002) compared to the DSN group.

## Conclusions

This study's findings reveal a significant and complex process of perceived social disengagement among caregivers describing social withdrawal and loss of social connections that they experienced as enduring. The results emphasize the need for early interventions that address the individual's needs but also address the caregiver's social and mental health to prevent perceived long-term social isolation.

## Introduction

Down Syndrome Regression Disorder (DSRD) is an acute or subacute neurocognitive regression in individuals with Down syndrome (DS), characterized by a loss of previously acquired cognitive, adaptive, and social skills [1]. DSRD has predominantly been observed in individuals with DS between the ages of 10 and 30. Symptoms include regression or loss of language, communication, cognition, and executive function, and can also include psychiatric manifestations, bradykinesia, catatonia, and rapid-onset insomnia. Regression results in a profound loss of ability to engage in activities of daily living, to the point where most individuals become dependent on their caregivers. This can cause caregivers to experience sleep disturbances, financial distress, and negative impacts to family dynamics and social relationships, among other challenges [2]. The time-intensive nature of care demands means that individuals with DSRD require significantly more support than they did prior to their onset of symptoms, with potential cascading negative effects on caregivers and their social relationships. While previously published literature has identified caregiver burden [2], dedicated exploration of the impact on social relationships of caregivers is unknown.

Previous research has revealed that parents of children with neurological conditions such as epilepsy, cerebral palsy, and Autism Spectrum Disorder (ASD), experience impacts to their social networks as well [3,4]. These conditions are associated with seizure, spasticity, and neurocognitive issues which are different than the largely neuropsychiatric symptoms reported in individuals with DSRD. The substantial care demands often leave caregivers with little opportunity to engage in and maintain friendships or participate in social activities, and others may struggle to adapt to the time requirements of caregiving. Over time, this strain can erode social connections and place additional stress on marital and partner relationships [5]. Additionally, caregivers have reported increased social exclusion and stigma, resulting in a significant decrease of their social relationships by both contacts and quality [6,7]. Similarly, research focused on caregivers of people with dementia and Alzheimer's disease (AD), showed reduced frequency of social activities and a limited ability to leave

home, causing social isolation [8,9]. With appropriate identification, interventional strategies have been identified in DS populations [10,11], making the need for identification significant.

The purpose of this study was to characterize caregiver-reported social impacts associated with caring for an individual with DSRD and compare these reported impacts with a heterogeneous caregiver group of individuals with DS and other neurological disorders (DSN) [12]. This study also characterized caregiver perceptions regarding changes in the quality of adult friendships, impact on social contacts, impact on spouse/partner, and social relationships. Understanding how the unique demands of DSRD caregiving may be associated with caregiver perceptions of reduced social support, is important in highlighting the overall care burden and mental health outcomes.

## Materials and methods

### Regulatory approval and data availability

This study was approved by the institutional review board at Children's Hospital Los Angeles and the University of Southern California (IRB number: CHLA-24–00184). Caregivers or guardians provided virtual consent, which involved a standard online information page that explained the study's purpose, expected duration, and potential risks and benefits, with a choice to participate. Anonymized data is available in the Dryad open data publishing platform with accession ID: 10.5061/dryad.cz8w9gjjg.

### Sample and data

This is a narrative burden-of-care study, not a network study and utilized a cross-sectional study design. This study had two separate groups of participants: 1) DSRD Cohort: This group consisted of caregivers of individuals with a confirmed or probable diagnosis of DSRD. They were recruited through a neurology clinic at Children's Hospital Los Angeles and a private Facebook support group for DSRD caregivers. The recruitment period lasted three weeks (July 6, 2024, to July 28, 2024). 2) DSN Cohort: This group was composed of caregivers of individuals with DS who also had other active neurological conditions, such as atlanto-axial instability, autism spectrum disorder, cerebrovascular disease, or epilepsy. Patients were required to have an active neurologic problem that was symptomatic at the time of the survey being completed. To be eligible for this study, caregivers in both cohorts had to be over 18 years old and generally knowledgeable about their loved one's condition.

After the survey, all DSRD caregiver responses were reviewed to ensure the reported symptoms aligned with the established criteria for a DSRD diagnosis. The survey directly included the 2022 international DSRD criteria, asking participants to report which symptoms were present at the onset of the condition [1]. A medical reviewer, who was not aware of any demographic information, conducted this review. This approach was intended to ensure accuracy and prevent data from individuals without DSRD from being included in the cohort. Caregivers whose loved ones did not meet at least three core DSRD symptoms were excluded from the study [1].

In the DSN group, the diagnosis of the neurological condition was based solely on caregiver self-report, and diagnosis by a physician was required in all cases.

### Measure of variables

Prior to release of the survey to prospective participants, the survey was validated for readability, comprehension and time to complete by a group of 10 caregivers. Feedback was collected in an open format and modifications for language, user-experience, and organization were made based on the feedback of these caregivers.

Caregivers voluntarily completed an anonymous online survey via REDCap (Appendix A in S1 File). The survey's initial page explained the consent process, and access to the questions was granted only after consent was provided. The survey gathered demographic and clinical information about the individual with DSRD or DSN (Appendix B in S2 File).

The survey included a section called the DSRD Caregiver Distress Survey (CDS) which featured four specific open-ended qualitative questions designed to capture the nuanced impact of caregiving on a caregiver's social relationships.

Thematic coding was applied to the responses from these four questions, with each question yielding five distinct codes, for a total of 20 codes.

To analyze these qualitative responses, a thematic coding process was utilized.

We analyzed caregiver free-text responses to four open-ended survey items addressing impacts of the individual's condition on adult friendships, marital/partner relationships, perceived social support, and perceived changes in social relationships. Caregivers were the unit of analysis. Responses were exported verbatim, de-identified, and segmented into "meaning units" (a sentence or clause expressing a single idea); a single response could contribute multiple meaning units. We conducted a semantic thematic analysis using a hybrid inductive–deductive coding strategy: an initial deductive code "start list" was derived from the survey domains (e.g., friendship disruption, partner strain, social support changes, social relationship and contact contraction), and inductive codes were added iteratively to capture unanticipated content (e.g., stigma-related withdrawal, time burden, role overload, health system navigation). Two coders independently reviewed and coded an initial subset of responses (15%) to develop a shared codebook specifying code definitions, inclusion/exclusion criteria, and exemplar quotations. The codebook was refined through iterative consensus meetings with a group of four authors until stable, after which the full dataset was coded with multiple codes permitted per meaning unit when appropriate. Discrepancies between coders were resolved by consensus discussion; if consensus could not be reached, the senior author adjudicated. Following coding, related codes were collated into higher-order themes through iterative review, with themes defined (scope and boundaries), checked against the full dataset, and refined to ensure internal coherence and distinctiveness across themes. To aid interpretability, the authors reported each theme with representative quotations; where frequencies are provided, they reflect the proportion of caregiver responses mentioning a code/theme and are intended as descriptive summaries rather than prevalence estimates. Finalized coded themes are referenced in Table 1.

All open-ended responses were coded using these themes, and the frequency of each code was tallied for both the DSRD and DSN groups. Where reported, code frequencies reflect the proportion of responses in which a theme was mentioned and are presented to aid comparison between cohorts rather than as population prevalence estimates.

The study did not use medical records; instead, it relied on caregiver-provided demographics and clinical information. To prevent multiple submissions, IP addresses were checked, but no cookies were used to track responses. The completion rate was calculated by dividing the number of completed surveys by the total number of consents obtained.

## Data analysis procedure

Descriptive statistics, including counts and percentages, were calculated to summarize the sample characteristics and key outcomes. To determine the statistical significance of differences in proportions between the two study groups, a Z-test for

**Table 1. Themes and corresponding subthemes of coding scheme.**

| Quality of Adult Friendships | Impact on Social Relationships | Spouse/Partner Impact | Perceived Shrinkage of Social World |
|---|---|---|---|
| Social Withdrawal and Isolation | Social Withdrawal & Isolation | Emotional and Interpersonal Strain | Time and Energy Constraints |
| Emotional Fatigue of Caregivers | Caregiving Constraints | Disagreements on Caregiving & Treatment | Friends' Discomfort or Inability to Relate |
| Social Disconnection and Exclusion | Behavioral Barriers & Stigma | Lack of Time and Intimacy | Reduced Invitations from Social Contacts |
| Negative Mental Health Impact | Loss of Community Participation and/or Support | Caregiving Stress-Related Relationship Deterioration | Shift to Condition-Specific Support Relationships |
| Difficulty Leaving Home or Including Loved One with DSRD | Decreased Social Contacts | Unequal Roles or Perceived Imbalance | Perceived Enduring Loss of Social Connections |

two population proportions was employed. This analysis yielded a Z-score and a corresponding two-tailed p-value for each outcome. A *p*-value of less than 0.05 was considered to be statistically significant, with values trending toward this threshold noted for further consideration. All *p*-values are reflected of adjusted analysis to control for age, ethnicity and race.

For the Quality of Adult Friendships item, the initial five coding categories were consolidated into three during post-hoc analysis to reduce redundancy and address conceptual overlap which existed on many responses. The three final categories were defined to comprehensively capture all participant responses, thereby improving clarity and interpretability of the data.

To evaluate whether caregiver distress differed with age, we modeled each caregiver distress outcome as a function of age (years) using logistic regression separately within the DSRD and DSN cohorts, reporting odds ratios (ORs) per 1-year increase in age with 95% confidence intervals. We additionally fit multivariable models adjusting for sex and symptom duration. Caregiver distress outcomes were defined as endorsement of CDS caregiver-impact items (e.g., worsening mental health, social withdrawal/isolation, diminished social support, partner strain), analyzed individually.

## Results

This study analyzed data from a total of 365 caregivers, with 228 in the DSRD cohort (62.5%) and 137 in the DSN cohort (37.5%). A flow diagram for the study completion, and subsequent qualitative responses is provided as Fig 1. Caregiver-reported clinical and demographic data regarding the cohorts is presented in Table 2. This data has been previously published in the initial assessment of caregiver burden in individuals with DSRD [2]. In the DSRD cohort, 141 (61.8%) were recruited from clinics and 87 (38.2%) were recruited from online groups compared to 94 (68.6%) and 43 (31.4%) in the DSN cohort, respectively (*p* = 0.19).

In the DSRD cohort, more than two-thirds presented altered mental status (71.1%), movement disorder (70.2%), language deficits (69.7%), psychiatric symptoms (67.1%), and insomnia (65.8%), with nearly half experiencing cognitive

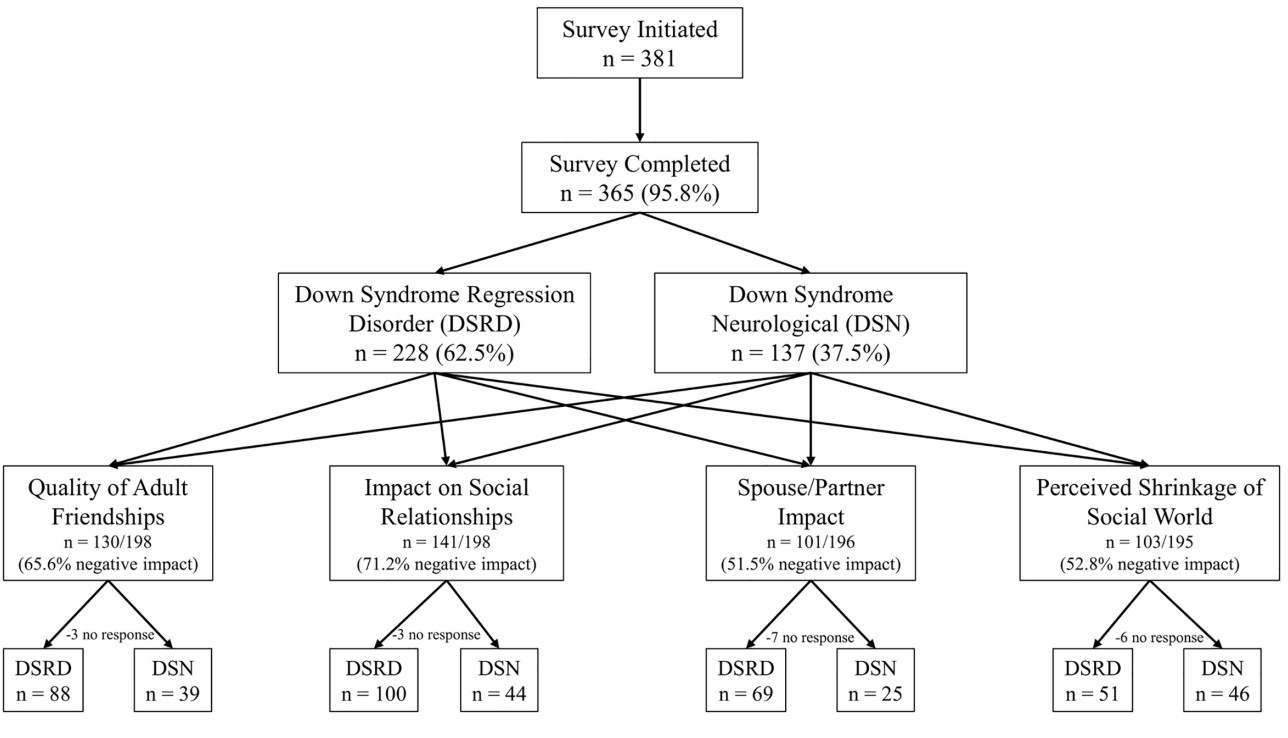

**Fig 1. Flow Diagram.**

**Table 2. Demographic and clinical characteristics by Down syndrome regression disorder and Down syndrome and other neurological conditions cohorts.**

| | DSRD | DSN | Total | p |
|---|---|---|---|---|
| | (n = 228) | (n = 137) | (n = 365) | |
| *Demographics of individual* | | | | |
| Age (years)* | 20.13 (7.38) | 16.05 (5.25) | 18.60 (6.94) | **<0.001** |
| Sex at birth | | | | 0.37 |
| Female | 111 (48.7%) | 60 (43.8%) | 171 (46.8%) | |
| Male | 117 (51.3%) | 77 (56.2%) | 194 (53.2%) | |
| Ethnicity | | | | 0.37 |
| Non-Hispanic | 168 (73.7%) | 95 (69.3%) | 263 (72.1%) | |
| Hispanic | 60 (26.3%) | 42 (30.7%) | 102 (27.9%) | |
| Race | | | | 0.78 |
| White | 170 (74.6%) | 100 (73.0%) | 270 (74.0%) | |
| Black/African American | 32 (14.0%) | 19 (13.9%) | 51 (14.0%) | |
| Asian | 17 (7.5%) | 13 (9.5%) | 30 (8.2%) | |
| Native American/Alaskan Native | 7 (3.1%) | 5 (3.6%) | 12 (3.3%) | |
| Hawaiian or Pacific Islander | 2 (0.9%) | 0 (0.0%) | 2 (0.5%) | |
| Duration of symptoms (years)ᵃ | 4.37 (3.80) | 4.59 (2.19) | 4.45 (3.29) | 0.54 |
| *Demographics of caregiver* | | | | |
| Relationship to individual | | | | 0.61 |
| Mother | 166 (72.8%) | 106 (77.4%) | 272 (74.5%) | |
| Father | 43 (18.9%) | 22 (16.1%) | 65 (17.8%) | |
| Guardian | 11 (4.8%) | 9 (6.7%) | 20 (5.5%) | |
| Age of caregiver (median, IQR) | 56.2 (6.7) | 54.9 (6.2) | 55.7 (6.5) | 0.53 |
| Ethnicity | | | | 0.47 |
| Non-Hispanic | 163 (71.5%) | 93 (67.8%) | 256 (70.1%) | |
| Hispanic | 65 (28.5%) | 44 (32.1%) | 109 (29.9%) | |
| Race | | | | 0.44 |
| White | 173 (75.9%) | 99 (72.2%) | 272 (74.5%) | |
| Black/African American | 32 (14.0%) | 19 (13.9%) | 51 (14.0%) | |
| Asian | 18 (7.9%) | 14 (10.2%) | 32 (8.8%) | |
| Native American/Alaskan Native | 5 (2.2%) | 5 (3.6%) | 10 (2.7%) | |
| Primary language at home non-English | 29 (12.7%) | 21 (15.3%) | 50 (13.7%) | 0.48 |
| *Disease characteristics of DSRD patients* | | | | |
| Movement disorder | 160 (70.2%) | | | |
| Altered mental status | 162 (71.1%) | | | |
| Cognitive decline | 112 (49.1%) | | | |
| Focal neurologic deficits/seizure | 11 (4.8%) | | | |
| Insomnia | 150 (65.8%) | | | |
| Language deficits | 159 (69.7%) | | | |
| Psychiatric symptoms | 153 (67.1%) | | | |
| Total symptoms | 4.98 (1.51) | | | |

*(Continued)*

**Table 2.** (Continued)

| | DSRD | DSN | Total | p |
|---|---|---|---|---|
| | (n = 228) | (n = 137) | (n = 365) | |
| *Disease Characteristics of DSN patients*γ | | | | |
| Atlanto-axial instability | | 11 (8.0%) | | |
| Autism spectrum disorder | | 69 (50.3%) | | |
| Cerebrovascular disease/stroke | | 15 (10.9%) | | |
| Epilepsy | | 42 (30.7%) | | |

Data are frequency (%) or mean (SD). *$p < 0.05$ (in bold font).

DSRD: Down syndrome regression disorder; DSN: Down syndrome with neurological disorders.

α: indicates years since the diagnosis of this condition; β: United States geographic regions per census criteria; γ: primary diagnosis related to what patient needs to see a neurologist for.

decline (49.1%) and only 4.8% having focal neurologic deficits/seizure. On average, individuals with DSRD reported five core "diagnostic" symptoms of the condition (SD = 1.5).

## Qualitative CDS item responses

In the DSRD group, individuals were, on average, older than those in the DSN group (20.1 years [SD = 7.4] vs. 16.1 years [SD = 5.3], $p < 0.001$), while the groups were comparable in terms of sex, ethnicity, race, and duration of symptoms (Table 2). Caregivers of individuals with DSRD reported greater burden than those caring for individuals with DSN on quantitative CDS item responses. Notably, caregivers of individuals with DSRD were more likely to report perceived declines in the quality of adult friendships (64.4% vs. 29.2%, $p < 0.001$), overall social relationships (71.1% vs. 34.3%, $p < 0.001$), and marital relationships (52.0% vs. 19.7%, $p < 0.001$) compared with DSN caregivers. Additionally, a greater proportion of DSRD caregivers reported a perceived shrinkage of social world (54.1% vs. 38.0%, $p = 0.01$). Furthermore, DSRD caregivers reported experiencing worsening mental health (77.5% vs. 21.2%, $p < 0.001$) compared to DSN caregivers (Table 3).

## Qualitative themes from open responses

Qualitative caregiver responses revealed significant impacts across multiple social domains (Fig 1). Out of 198 total responses regarding the Quality of Adult Friendships, 130 (65.6%) reported a negative impact. For Social Relationships Impacted, 141 out of 198 responses (71.2%) indicated a negative impact. Regarding Spouse/Partner Impact, 101 out of 196 responses (51.5%) were negatively impacted. Finally, for Perceived Shrinkage of Social World, 103 of 195 responses (52.8%) reported a negative impact. The DSRD and DSN self-reported responses are summarized in Table 4.

For evaluation of adult friendships, descriptive quotations were provided by 88 participants in the DSRD group and 39 participants in the DSN group. A significant difference was found in the impact on the quality of adult friendships (Fig 2a). DSRD caregivers reported a higher percentage of social withdrawal and isolation (43.2% vs. 17.9%, $p = 0.01$, [0.1, 0.4]) compared to DSN caregivers. Conversely, emotional fatigue was significantly higher among DSN caregivers than DSRD caregivers (61.5% vs. 31.8%, $p = 0.01$, [−0.5, −0.1]). There was no significant difference between groups in negative mental health impact (25.0% vs. 20.5%, $p = 0.58$, [−0.1, 0.2]).

Qualitative responses regarding social relationships were provided by 100 DSRD participants and 44 DSN participants. Caregivers reported impacts on various aspects of their social interactions (Fig 2b). DSRD caregivers reported a significantly higher percentage of "Loss of Community Participation and/or Support" (20.0% vs. 4.5%, $p = 0.02$, [0.05, 0.25]), whereas DSN caregivers reported a significantly higher percentage of "Caregiving Constraints" (27.1% vs. 56.8%, $p < 0.001$, [−0.48, −0.14]).

In total, 69 participants in the DSRD group and 25 in the DSN group provided descriptive quotations regarding the impact of diagnosis on spouse or partner relationships. The analysis of spouse and partner impact revealed insignificant

**Table 3. Responses of CDS items by DSRD and DSN caregiver groups.**

| | DSRD | DSN | Total | p |
|---|---|---|---|---|
| | (n = 228) | (n = 137) | (n = 365) | |
| *Social contacts and relationships* | | | | |
| Quality of adult friendships impacted | | | | |
| Positive or no change | 80 (35.6%) | 97 (70.8%) | 177 (48.9%) | **<0.001** |
| Negative change | 145 (64.4%) | 40 (29.2%) | 185 (51.1%) | |
| Social relationships impacted | | | | |
| Positive or no change | 65 (28.9%) | 90 (65.7%) | 155 (42.8%) | **<0.001** |
| Negative change | 160 (71.1%) | 47 (34.3%) | 207 (57.2%) | |
| Marriage impacted | | | | |
| Positive or no change | 107 (48.0%) | 110 (80.3%) | 217 (60.3%) | **<0.001** |
| Negative change | 116 (52.0%) | 27 (19.7%) | 143 (39.7%) | |
| Perceived shrinkage of social world | | | | |
| No or expanded social world | 102 (45.9%) | 85 (62.0%) | 187 (52.1%) | **0.01** |
| Yes | 120 (54.1%) | 52 (38.0%) | 172 (47.9%) | |
| *Mental health* | | | | |
| Worsened mental health | | | | |
| No | 50 (22.5%) | 108 (78.8%) | 158 (44.0%) | **<0.001** |
| Yes | 172 (77.5%) | 29 (21.2%) | 201 (56.0%) | |
| Frustration/anger towards loved one | | | | |
| No | 104 (46.2%) | 126 (92.0%) | 230 (63.5%) | **<0.001** |
| Yes | 121 (53.8%) | 11 (8.0%) | 132 (36.5%) | |
| Fears about child's future or family's future | | | | |
| No | 13 (5.8%) | 44 (32.1%) | 57 (15.8%) | **<0.001** |
| Yes | 211 (94.2%) | 93 (67.9%) | 304 (84.2%) | |
| Fears about misdiagnosis | | | | |
| No | 71 (32.1%) | 113 (82.5%) | 184 (51.4%) | **<0.001** |
| Yes | 150 (67.9%) | 24 (17.5%) | 174 (48.6%) | |
| Fears about access to treatment | | | | |
| No | 34 (15.4%) | 75 (54.7%) | 109 (30.4%) | **<0.001** |
| Yes | 187 (84.6%) | 62 (45.3%) | 249 (69.6%) | |

Data are frequency (%). *$p < 0.05$ shown in bold. DSRD: Down syndrome regression disorder; DSN: Down syndrome with neurological disorders; CDS: Caregiver distress survey. The frequency and percent of incomplete variables: Quality of adult friendships impacted (n = 3, 0.8%); Social relationships impacted (n = 3, 0.8%); Marriage impacted (n = 5, 1.4%); Perceived shrinkage of social world (n = 6, 1.6%); Worsened mental health (n = 6, 1.6%); Frustration/anger towards loved one (n = 3, 0.8%); Fears about child's future or family's future (n = 4, 1.1%); Fears about misdiagnosis (n = 7, 1.9%); Fears about access to treatment (n = 7, 1.9%).

findings (Fig 3a). All percentages are reported as (DSRD vs. DSN) in this section. All subthemes, including "Emotional and Interpersonal Strain" (20.29% vs. 8.00%, p = 0.16, [−0.02. 0.27]), "Disagreements on Caregiving & Treatment" (15.94% vs. 24.00%, p = 0.37, [−0.3, 0.1]), "Lack of Time and Intimacy" (27.54% vs. 40.00%, p = 0.25, [−0.3, 01), "Caregiving Stress-Related Relationship Deterioration" (28.98% vs 20.00%, p = 0.38, [−0.1, 0.3]), and "Unequal Roles or Perceived Imbalance" (7.25% vs. 8.00%, p = 0.90, [−0.1. 0.1]), did not meet the threshold for statistical significance.

Regarding perceived social contacts, 51 DSRD caregivers and 46 DSN caregivers provided written responses. Caregiver-reported results indicated a significantly higher percentage of DSRD caregivers reported a 'Perceived Enduring Loss of Social Connections' (35.3% vs. 8.7%, p = 0.01, [0.1, 0.4]) compared to the DSN group (Fig 2b). However, the DSN group reported a significantly higher percentage of "Time and Energy Constraints" compared to DSRD caregivers (56.5% vs. 15.7%, p < 0.001, [−0.6, −0.2]). "Reduced Invitations from Social Contacts" was more prevalent in the DSRD group (19.6%

**Table 4. Qualitative responses by DSRD and DSN caregiver groups.**

| | DSRD | DSN | p |
|---|---|---|---|
| *Quality of Adult Friendships* | (n = 88) | (n = 39) | |
| Social Withdrawal and Isolation | 38 (43.2%) | 7 (17.9%) | 0.01 |
| Emotional Fatigue of Caregivers | 28 (31.8%) | 24 (61.5%) | **0.01** |
| Negative Mental Health Impact | 22 (25.0%) | 8 (20.5%) | 0.58 |
| *Social Relationships Impacted* | (n = 100) | (n = 44) | |
| Social Withdrawal and Isolation | 16 (16.0%) | 7 (15.9%) | 0.99 |
| Caregiving Constraints | 26 (26.0%) | 25 (56.8%) | **<0.001** |
| Behavioral Barriers and Stigma | 16 (16.0%) | 5 (11.4%) | 0.47 |
| Loss of Community Participation and/or Support | 20 (20.0%) | 2 (4.5%) | **0.02** |
| Decreased Social Contacts | 22 (22.0%) | 5 (11.4%) | 0.13 |
| *Spouse and Partner Impact* | (n = 69) | (n = 25) | |
| Emotional and Interpersonal Strain | 14 (20.3%) | 2 (8.0%) | 0.16 |
| Disagreements on Caregiving & Treatment | 11 (15.9%) | 6 (24.0%) | 0.37 |
| Lack of Time and Intimacy | 19 (27.5%) | 10 (40.0%) | 0.25 |
| Caregiving Stress-Related Relationship Deterioration | 20 (29.0%) | 5 (20.0%) | 0.38 |
| Unequal Roles or Perceived Imbalance | 5 (7.2%) | 2 (8.0%) | 0.90 |
| *Perceived Shrinkage of Social World* | (n = 51) | (n = 46) | |
| Time and Energy Constraints | 8 (15.7%) | 26 (56.5%) | **<0.001** |
| Friends' Discomfort or Inability to Relate | 7 (13.7%) | 3 (6.5%) | 0.24 |
| Reduced Invitations from Social Contacts | 10 (19.6%) | 6 (13.0%0 | 0.38 |
| Shift to Condition-Specific Support Relationships | 8 (15.7%) | 7 (15.2%) | 0.95 |
| Perceived Enduring Loss of Social Connections | 18 (35.3%) | 4 (8.7%) | **0.01** |

Data are frequency (%). $^*p < 0.05$ shown in bold. DSRD: Down syndrome regression disorder; DSN: Down syndrome with neurological disorders.

vs. 8.7%, $p = 0.38$, [−0.1, 0.2]). Caregiver distress did not worsen with increasing age in either cohort. In the DSRD group, age was not associated with higher odds of endorsing worsening caregiver outcomes (per 1-year increase: OR 1.01; 95% CI 0.98–1.04; $p = 0.58$). Similarly, in the DSN group, age was not associated with worsening caregiver outcomes (per 1-year increase: OR 0.99; 95% CI 0.95–1.03; $p = 0.66$). Results were unchanged after adjustment for sex and symptom duration (DSRD adjusted OR 1.00, 95% CI 0.97–1.03, $p = 0.91$; DSN adjusted OR 0.98, 95% CI 0.94–1.02, $p = 0.31$).

A table of descriptive quotations illustrating these themes and providing a deeper qualitative perspective is included in Table 5. Additionally, for each theme, Figs 2 and 3 illustrate a visual representation of the percentage of DSRD vs. DSN caregivers per subtheme.

## Discussion

In this cross-sectional comparison of caregivers of individuals with DSRD versus DSN, we found consistent caregiver-reported perceptions of social strain and reduced social connectedness across multiple domains, with the clearest between-group differences involving social withdrawal, reduced community participation, and perceived enduring loss of connections. Prior caregiver studies in neurodevelopmental and neurologic conditions describe isolation, stigma, and reduced participation; our contribution is describing caregiver-reported social impacts in the DSRD cohort relative to a heterogeneous DSN comparison group. While between-group differences were observed in reported community participation and reported loss of social connections, these findings should be interpreted cautiously given the self-reported nature of outcomes and the heterogeneity of neurologic diagnoses within the DSN comparator, which limits disorder-specific inference [12]. While these differences are statistically significant within this sample, they represent subjective narratives

a)

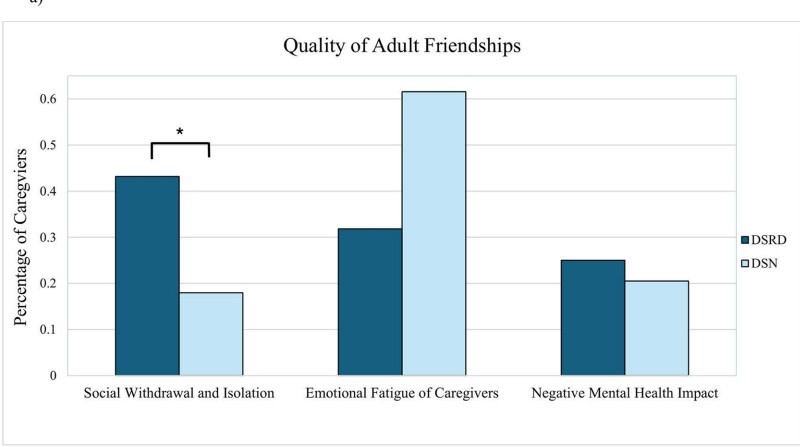

b)

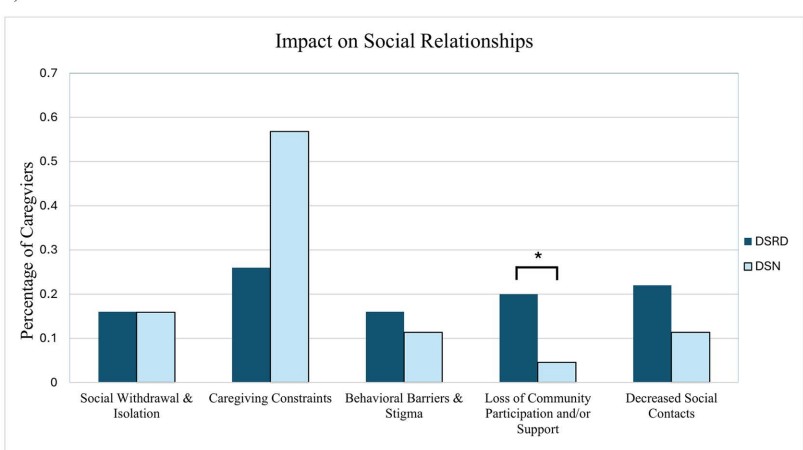

**Fig 2. Percentage of Down syndrome regression disorder vs. Down syndrome with neurological condition caregivers per subtheme in adult friendships and social relationships.** Panel (a) presents the "Quality of Adult Friendships" subthemes, showing the percentage of caregivers for DSRD (dark blue bars) and DSN (light blue bars) experiencing the subthemes: "Social Withdrawal and Isolation," "Emotional Fatigue of Caregivers," and "Negative Mental Health Impact." Panel (b) illustrates the "Social Relationships Impacted" subthemes, detailing the percentage of caregivers affected by "Social Withdrawal & Isolation," "Caregiving Constraints," "Behavioral Barriers & Stigma," "Loss of Community Participation and/or Support," and "Decreased Social Contacts." Asterisks (*) indicate statistically significant differences between groups (p < 0.05).

and should not be used to infer structural network differences between disorders. Caregivers described caregiving demands as associated with reduced social engagement and fewer opportunities to sustain friendships and community participation, which, when repeatedly reinforced, were perceived as progressive contact/relationship loss. Together, these findings suggest a coherent pattern in which caregiving constraints, social withdrawal, and reduced community participation co-occur with perceptions of perceived enduring loss of social connections, while acknowledging that longitudinal data are needed to assess persistence over time. Clinically, these results support routine screening for caregiver isolation and proactive linkage to respite, peer support, and navigation resources; approaches that may help preserve social connectedness early in the DSRD course [10,11].

The most statistically significant findings in this study centered on caregivers' descriptions of loss of social connections that they experienced as enduring. Specifically, caregivers of individuals with DSRD reported increased social withdrawal and isolation and a loss of community participation and support, suggesting a disengagement from social functions and

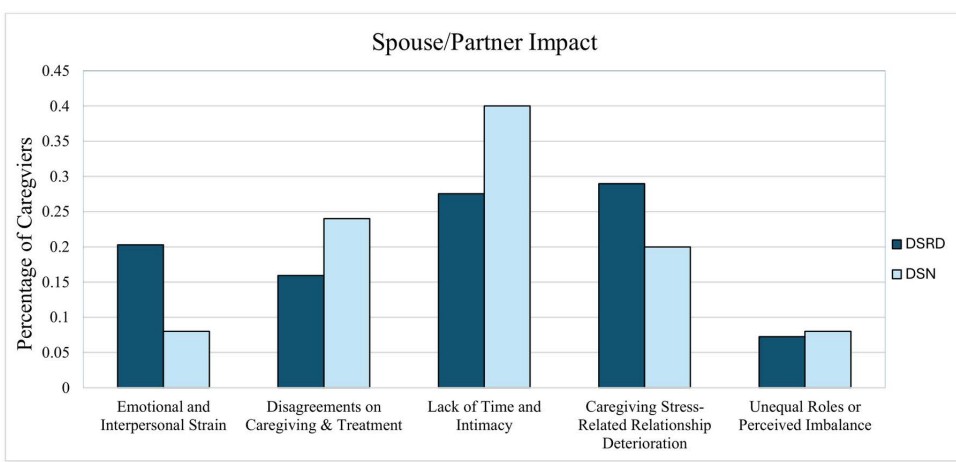

a)

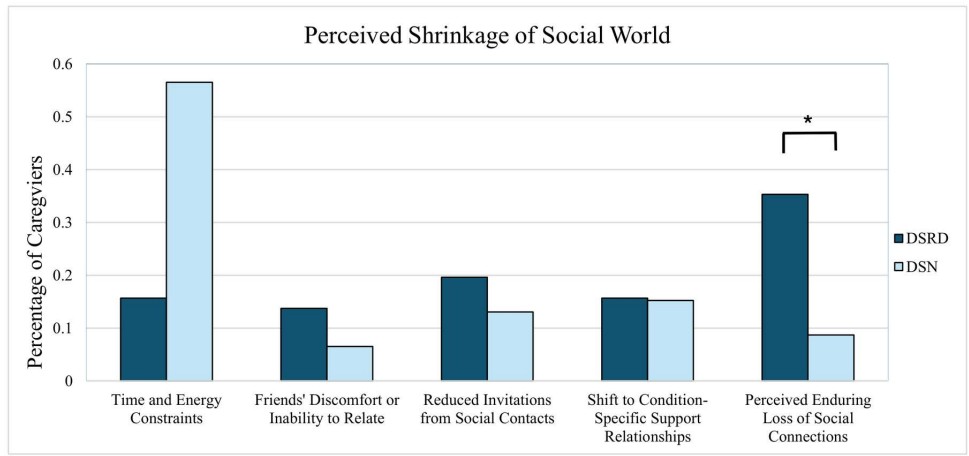

b)

**Fig 3. Percentage of Down syndrome regression disorder vs. Down syndrome with neurological condition caregivers per subtheme in spouse and partner impact and perceived shrinkage of social world.** Panel (a) shows the percentage of caregivers of DSRD vs. DSN groups for subthemes within the "Spouse/Partner Impact" theme, including "Emotional and Interpersonal Strain," "Disagreements on Caregiving & Treatment," "Lack of Time and Intimacy," "Caregiving Stress-Related Relationship Deterioration," and "Unequal Roles or Perceived Imbalance." Panel (b) shows the percentages for subthemes within the "Perceived Shrinkage of Social World" theme, including "Time and Energy Constraints," "Friends' Discomfort or Inability to Relate," "Reduced Invitations from Social Contacts," "Shift to Condition-Specific Support Relationships," and "Perceived Enduring Loss of Social Connections." Asterisks (*) indicates statistically significant differences between groups (p < 0.05).

interactions within their communities. These challenges align with a broader finding that caregiving constraints, particularly a lack of time and energy, affect a caregiver's ability to maintain their social relationships. This diminished capacity for social engagement is not merely a temporary inconvenience but was described by caregivers as contributing to an enduring loss of social connections. More broadly, these results suggest that the challenges associated with caregiving lead to significant consequences that may alter the caregiver's social experiences and perceived support. These data have never been reported in individuals with DSRD and a question that has emerged is what the longitudinal impact of these findings is based on clinician-based diagnosis and treatment. Future studies evaluating this effect could inform the need for early identification of caregiver social impact and potential intervention.

**Table 5. Qualitative findings on the social relationships of DSRD caregivers.**

| Theme | Subtheme | Representative Quote |
|---|---|---|
| **Quality of Adult Friendships** | Social Withdrawal and Isolation | "I am almost completely isolated and unable to have social interaction outside of my home. Most don't understand why I can't just "meet up" for coffee." |
| | Emotional Fatigue of Caregivers | "Can't spontaneously take part in activities… Wondering if everything is ok at home. Never leaves you." |
| | | "I don't have the time to spend with friends. All I talk about is my son's DSRD." |
| | Negative Mental Health Impact | "My friendships are deteriorating. I find it hard to share in any real, honest way what I'm dealing with with my son's DSRD. I feel like they don't understand. I feel isolated … We're struggling, but I don't want people to know." |
| **Impact on Social Relationships** | Social Withdrawal & Isolation | "Zero social networks remain" |
| | | "Very isolated now" |
| | Caregiving Constraints | "No time or emotional energy for anything at all outside of getting through every day" |
| | Behavioral Barriers & Stigma | "Friends avoid because of strange behavior. Family stopped inviting. No social invites" |
| | | "People don't understand, even parents in the Down syndrome groups. We sort of feel ostracized & pitied" |
| | Loss of Community Participation and/or Support | "We have been unable to leave the house to attend church—a significant loss to our entire family" |
| | | "Fired from my job of 18 years" |
| | Decreased Social Contacts | "Most "friends" disappeared or moved on without us" |
| | | "Community? What's that?" |
| **Spousal and Partner Impact** | Emotional and Interpersonal Strain | "We argue and are resentful of one another and have zero alone time." |
| | | "Spouse is not supportive. He's resentful of the time I spend caring for my son" |
| | Disagreements on Caregiving & Treatment | "Increased arguments. Less patience with each other. Differences in how to manage our son" |
| | Lack of Time and Intimacy | "Stress, worry, frustration and sleep deprivation cause arguments. Sleeping apart and lack of time has greatly affected our sex life" |
| | Caregiving Stress-Related Relationship Deterioration | "We are on the verge of getting a divorce. He thinks we should be doing more but doesn't help at all" |
| | | "Divorce — husband (stepdad) got frustrated with symptoms, stress, and financial burden. Even though I did 99% of the work, he left and filed for divorce" |
| | Unequal Roles or Perceived Imbalance | "My husband handles her better, which is frustrating because I'm with her more" |
| **Perceived Shrinkage of Social World** | Time and Energy Constraints | "I have work and my daughter. That's it." |
| | | "I have no energy for friends." |
| | Friends' Discomfort or Inability to Relate | "People don't know what to say and so they say nothing." |
| | | "The amount of interaction has significantly declined. Nobody knows how to relate." |
| | Reduced Invitations from Social Contacts | "We had to say no too many times. We are not on people's minds anymore." |
| | Shift to Condition-Specific Support Relationships | "We do not have much of a social network anymore outside of parents who get it." |
| | | "All of our friends are from the DSRD community now." |
| | Perceived Enduring Loss of Social Connections | "It's gone. All of it." |
| | | "I lost everyone." |

Comparisons between caregivers of individuals with DSRD and those in the DSN group should be interpreted cautiously. While similar themes of social isolation, decreased community participation, and relationship strain are reported by caregivers of children with other neurological disorders, in this respondent sample, caregivers in the DSRD group more frequently perceived constraints on social engagement than caregivers in the DSN group. For instance, studies on children with cerebral palsy have shown significantly lower community and home participation, with caregivers identifying

physical, cognitive, and social barriers [13]. These limitations include a lack of transportation, assistive services, and financial support as key factors [13]. Likewise, research on caregivers in the broader DSN population, such as mothers of children with various long-term care needs, highlights similar challenges including social isolation and distance from friends stemming from a lack of knowledgeable support systems and gaps in respite services [14]. However, while these studies describe significant barriers to social engagement, they do not typically emphasize the permanence of the social loss. This distinction is crucial because although caregivers across many conditions in the DSN group face obstacles to community participation, DSRD caregivers more frequently reported social losses that they experienced as enduring; however, persistence and temporality cannot be determined from this cross-sectional survey. This degree of perceived enduring social loss may reflect higher caregiving burden among DSRD caregivers in this respondent sample. Such impacts of DSRD are not only limited to caregiving tasks, but also cascade into the caregiver's social, relational, and emotional life. Thus, interventions that support the individual's medical needs while also actively protecting the caregiver's social and mental health are urgently needed.

Higher levels of stress are associated with the care of children and adults with DS [15,16]. Caregivers more frequently reported a perceived shrinkage of the social world in the DSRD cohort, primarily driven by a time and energy constraints and reduced invitations from existing social contacts. It is the author's hypothesis that this pattern is a negative feedback loop wherein higher levels of care for loved ones with DSRD leads to decreased time and ability to engage socially which subsequently leads to decreased invitations for engagement from the existing social relationships. This pattern of decreased time to engage in social activity has been previously reported in individuals with DS [17–19] and the overall negative feedback loop of social isolated is established in caregivers of those with chronic neuropsychiatric disease [20–22]. We cannot attribute perceived caregiver social changes to any single factor in this cross-sectional design as prior work on social relationships being "static" in people with DS may not generalize to caregiver relationships [23].

While not statistically significant, the impact on spousal and partner relationships was notable and suggested a potential effect, indicating that the burdens of caregiving for a loved one with DSRD may also strain marriages. Literature on spousal impact in other neurological diseases have reported significant strain on marital relationships, leading to reduced couple time, increased conflict from unequal caregiving responsibilities, and high stress driven by stigma, unpredictable medical or behavioral needs, and fatigue [5,24,25]. For example, in a study of parents of children with ASD, behaviors consistent with autism were directly linked to parenting stress and conflict, which, in turn, predicted lower marital love and higher conflict between the couple [5]. Similarly, caregivers of children with epilepsy reported that the unpredictability of seizures and medical management disagreements strained their marriages [25]. Many of these concerns and themes were shared amongst the DSRD caregivers. Overall, the existing literature supports lower marital satisfaction, diminished cohesion in decision making, and in some cases, elevated risk of divorce compared with parents of typically developing children. While these challenges are most commonly reported, it is important to note that some couples also described increased resilience and strengthened bonds through their shared caregiving experience, potentially explaining the lack of significance of negative impact in this study [26].

This study has several limitations which must be considered when interpreting the data. First, reliance on caregiver self-report introduces the possibility of recall bias and social desirability bias and limits diagnostic verification, particularly in the DSN group. Because the DSN comparator is heterogeneous and diagnoses/outcomes were not clinically adjudicated within this study, between-group differences should be interpreted as differences in caregiver-reported experiences among respondents rather than disorder-specific effects attributable uniquely to DSRD. Importantly, we did not directly measure social network structure (size, density, or dynamics) using formal social network analysis; findings therefore reflect caregiver perceptions and narratives rather than objectively observed network properties or causal change. In addition, the tool was partially validated by caregivers for readability and clarity although the authors acknowledge that the survey has not yet undergone full psychometric validation (e.g., factor structure and test–retest reliability in independent samples). Second, the DSRD cohort included participants recruited from a disease-specific online support group, which

may overrepresent caregivers experiencing higher distress or social withdrawal who are specifically looking for support, reducing generalizability. In addition, this study is limited by its cross-sectional design as social impacts (and caregiver burden) may change over time. Further, treatment data was not collected due to the lack of ability to verify interventions, timing of interventions, and the lack of ability to adjust for regional therapeutic offering differences. The authors believe that successful treatment of DSRD would improve QOL and this represents a future area of study for the team. In addition, while the thematic coding process was consensus-based and methodologically rigorous, qualitative analysis remains inherently interpretive and may not fully capture the complexity of caregiver experiences. Further, the narrower nature of the coding may have oversimplified themes or omitted other areas of social support or strain. In addition, some language such as the use of the term "permanent" or "loss of community participation" can be hard to interpret both for those completing the survey and the researchers coding the data. Additionally, the DSN comparison group was intentionally heterogeneous, encompassing various neurologic conditions; this limits direct comparability and constrains disorder-specific inference, as observed differences may reflect differences in diagnosis mix, comorbidity burden, severity, and caregiver context rather than DSRD itself. This is further complicated by this study's lack of ability to clinically confirm diagnoses. Demographic factors such as socioeconomic status, caregiver education level, family structure and geographic location were not uniformly collected, further limiting generalizability. The authors feel strongly that certain factors such as socioeconomic status may be important drivers of perceived shrinkage of the social world and warrant exploration in the future as discrepancies with primary caregivers not having financial resources to obtain additional care support. Additionally, the cross-sectional design and lack of longitudinal follow-up preclude assessment of causality or temporal dynamics; accordingly, findings should be interpreted as associative caregiver reports rather than evidence of causal or progressive disorder-specific social relationship change. Future research should address these limitations by incorporating prospective designs, validated clinical assessments, and broader sociodemographic sampling to ensure more representative and generalizable conclusions.

## Conclusion

### Theoretical implications

This study advances the DSRD caregiving literature by focusing on caregiver social relationships (not only global burden). Across domains (friendships, partner relationships, social contacts), caregivers described perceived reductions in social connectedness and participation. By comparing caregiver reports in the DSRD cohort with those from a heterogeneous DSN group using a shared framework, we provide initial descriptive evidence of differential perceived social impacts; however, disorder-specific inference will require longitudinal studies with more diagnostically homogeneous comparators and clinically verified diagnoses.

### Health policy implications based on incentive systems for caregiver social relationships

These data support treating caregiver connectedness as a health-relevant resource that systems can protect or erode. Incentive-aligned policies could include 1) health system protocols (screening for caregiver isolation; referral to respite, navigation, and mental health supports, 2) clinic-embedded peer supports); payer coverage for care coordination, respite, and 3) structured caregiver programs, and employer/community supports (flexible scheduling, protected leave, accessible respite).

### Future research

Logical next steps are longitudinal, multi-site studies with stronger diagnostic characterization, detailed caregiver context, validated social-network measures plus interviews, and analyses of moderators (treatment course, clinical response, respite access). Longitudinal study design will also afford the ability to track impacts over treatment periods which may

yield unique fluctuations and assess if self-reported perceptions of loss are as enduring as reported in this study. Further, caregiver intervention studies can test whether incentive-aligned support can prevent relationship loss and downstream caregiver distress.

## Supporting information

**S1 File. Appendix A.** Survey, Part A.
(PDF)

**S2 File. Appendix B.** Survey, Part B.
(PDF)

## Acknowledgments

The authors would like to thank the families and caregivers who participated in this survey.

## Author contributions

**Conceptualization:** Katherine N. Chow, Eileen A. Quinn, Jonathan D. Santoro.

**Data curation:** Katherine N. Chow, Lilia Kazerooni, Maeve C. Lucas, Samuel T. Otey, Mariam M. Yousuf, Jonathan D. Santoro.

**Formal analysis:** Katherine N. Chow, Lilia Kazerooni, Mariam M. Yousuf, Jonathan D. Santoro.

**Investigation:** Katherine N. Chow, Jonathan D. Santoro.

**Methodology:** Katherine N. Chow, Maeve C. Lucas, Eileen A. Quinn, Jonathan D. Santoro.

**Project administration:** Samuel T. Otey, Ruth Brown, Jonathan D. Santoro.

**Resources:** Jonathan D. Santoro.

**Supervision:** Ruth Brown, Eileen A. Quinn, Jonathan D. Santoro.

**Validation:** Mariam M. Yousuf, Ruth Brown, Jonathan D. Santoro.

**Visualization:** Katherine N. Chow, Maeve C. Lucas, Mariam M. Yousuf, Ruth Brown, Jonathan D. Santoro.

**Writing – original draft:** Katherine N. Chow, Lilia Kazerooni, Maeve C. Lucas, Samuel T. Otey, Mariam M. Yousuf, Ruth Brown, Eileen A. Quinn, Jonathan D. Santoro.

**Writing – review & editing:** Katherine N. Chow, Lilia Kazerooni, Maeve C. Lucas, Samuel T. Otey, Mariam M. Yousuf, Ruth Brown, Eileen A. Quinn, Jonathan D. Santoro.

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
