## [Decision Letter · Decision Letter 0]

3 Dec 2025

Dear Dr. Santoro,

We look forward to receiving your revised manuscript.

Kind regards,

Inge Roggen, M.D., Ph.D.

Academic Editor

PLOS ONE

Journal Requirements:

Additional Editor Comments:

Thank you for the opportunity to read on this important subject, that needs more awareness, however, there are several sound remarks from the reviewers we would like to see addressed. They are summarized below.

Reviewers' comments:

Reviewer's Responses to Questions

**Comments to the Author**

1. Is the manuscript technically sound, and do the data support the conclusions?

Reviewer #1: Partly

Reviewer #2: Partly

Reviewer #3: Partly

2. Has the statistical analysis been performed appropriately and rigorously?

Reviewer #1: I Don't Know

Reviewer #2: Yes

Reviewer #3: Yes

3. Have the authors made all data underlying the findings in their manuscript fully available?

Reviewer #1: Yes

Reviewer #2: Yes

Reviewer #3: No

4. Is the manuscript presented in an intelligible fashion and written in standard English?

Reviewer #1: Yes

Reviewer #2: Yes

Reviewer #3: Yes

Reviewer #1: The authors deserve credit for highlighting the challenges faced by caregivers of people with DSRD. This is an important contribution of the article.

In the section entitled 'Limitations', the authors point out several weaknesses in the methodology, including the possibility of recall and selection biases, the lack of recording of demographic factors such as education and socioeconomic status, and the lack of longitudinal follow-up.

However, I have a few comments.

Participants were recruited from a neurology clinic and a Facebook support group for DSRD caregivers. However, the authors do not specify how many participants came from each group. It can be assumed that the DSRD diagnosis in the neurology clinic group is correct. This is more difficult to assume for the other group, as the survey contains a number of difficult medical terms, and participants were not provided with guidance by one of the researchers when completing it. This could be a potential source of inaccuracy. It is also unclear whether the survey was validated in advance, which could also be a potential source of inaccuracy.

The authors state that they collected demographic and clinical information via the survey. However, these questions are not included in Appendix A. Table 2 lists the mean age, gender, ethnicity, race, and mean duration of symptoms. However, in the Results section, no distinction is made according to ethnicity or race regarding the impact on caregivers. Thus, the significance of mentioning this in Table 2 is unclear.

I also miss a mention of how many of the DSRD group have already been treated and what the outcome was. It would have been interesting to analyze whether that would make a difference to the QOL of the caregivers.

It is also never mentioned who the caregiver is who completes the survey.

I have another question about Table 2. The mean duration of symptoms in the DSN group is 4.59 years. This surprises me because the authors describe these individuals as young people with ASD, epilepsy, or moyamoya vasculopathy. The text does not specify how many people in the DSN group have each diagnosis. Although the DSN group is younger on average (mean age of 16.05 years), I find it strange that the mean symptom duration is only 4.59 years. A diagnosis of ASD and epilepsy in children is often made at a younger age. Perhaps the authors should verify this. I also wonder how meaningful the comparison between the DSRD and DSN groups is.

The social media quotes are both instructive and moving.

Despite its methodological limitations, the article demonstrates that caring for someone with DSDR significantly impacts the social lives of caregivers. However, I have difficulty with the word "permanent" when it comes to social ties. I am not a native English speaker, so perhaps I have misunderstood the meaning of "permanent." To me, it means "lasting forever," which this cross-sectional study with imperfect methodology cannot demonstrate. Perhaps the authors mean something else; I would like to know what that is.

Some minor remarks:

- It appears that the text on lines 116-117 is repeated on lines 118-119.

- Line 179 states: "The DSRD and DSN objective responses are summarized in Table 4." Does "objective responses" mean the same as "descriptive quotations," as written on line 183? If so, it would be better to replace "objective responses" with "descriptive quotations" in the sentence.

- The figures are difficult to read.

Reviewer #2: Impact of Down Syndrome Regression Disorder on Caregiver Social Networks

The topics of this paper are interesting, though well known. The structure and content must be revised, and results have to be better explained by authors before to be reconsidered.

Title, clarify that is a case study on Children's Hospital Los Angeles and period under study.

Abstract has to clarify the goal, and specific health and social implications to face DSRD.

Introduction has to better clarify the research questions of this study, indicating the gap presents in literature that this study endeavors to cover, and provide more theoretical background. Clarify differences with this study cited (Caregiver burden and familial impact in Down Syndrome Regression Disorder. Orphanet J Rare Dis 20, 126. https://doi.org/10.1186/s13023-025-03644-0).After that authors can focus on the topics of this study to provide a correct analysis for fruitful discussion. (Please refer to the suggested readings, all of which should be thoroughly reviewed and incorporated into your text.).

The methods of this study are not clear. The section of Materials and methods must be re-structured with the following three sections and same order:

• Sample and data

• Measures of variables

• Data analysis procedure.

Insert a flow chart to clarify the logic of the study design.

Authors have to avoid a lot of subheadings that create fragmentation and confusion. If necessary, you can use bullet points (same comments for section of results and all sections).

Results.

The study employed a cross-sectional methodology, capturing caregiver experiences at a single point in time. This design limits the ability to establish causality or to track how caregiver social networks evolve over the course of the disorder. DSRD is often characterized by acute or subacute regression, and the trajectory of caregiver distress may fluctuate depending on disease progression, access to interventions, or changes in family circumstances. A longitudinal approach would have provided richer insights into the dynamic nature of social withdrawal and the potential for recovery or adaptation. Participants were recruited from a neurology clinic and a DSRD Facebook support group. While this strategy ensured access to a relatively large sample, it introduces potential bias. Caregivers active in online support groups may represent those who are more socially engaged or more distressed, skewing the results. Similarly, caregivers recruited from clinical settings may differ in socioeconomic status, healthcare access, or severity of the child’s condition compared to those not seeking specialized care. This recruitment strategy may therefore limit the generalizability of findings to the broader population of caregivers. The study relied exclusively on caregiver self-reports through the DSRD Caregiver Distress Survey (CDS). Self-reported data are inherently vulnerable to recall bias, social desirability bias, and subjective interpretation of questions. For example, caregivers may underreport positive aspects of their social networks or overemphasize negative experiences due to heightened emotional distress. Without triangulation from objective measures—such as social network mapping, behavioral observations, or longitudinal tracking—the findings remain dependent on subjective perceptions. Although the CDS included four open-ended questions, the scope was relatively narrow, focusing on adult friendships, social networks, spouse/partner impact, and social network size. Important dimensions of caregiver social functioning—such as workplace relationships, extended family dynamics, or participation in religious or cultural communities—were not explicitly explored. This omission may have overlooked significant areas of social support or strain that contribute to caregiver well-being. Moreover, the thematic coding process, while systematic, involved reducing rich qualitative narratives into frequency counts. This conversion risks oversimplifying the complexity of caregiver experiences. Nuanced differences in the depth, intensity, or context of social withdrawal may have been lost in translation. For example, the meaning of “loss of community participation” may vary widely across cultural or geographic contexts, but such distinctions were not captured in the quantitative analysis. The study did not provide detailed demographic breakdowns of participants, such as age, gender, ethnicity, socioeconomic status, or geographic location. These factors are critical in shaping both caregiving experiences and social networks. Without accounting for diversity, the findings may disproportionately reflect the experiences of certain groups while neglecting others. For instance, rural caregivers may face different challenges than urban caregivers, and cultural norms may influence the extent to which social withdrawal is perceived or reported.

Although the survey included questions on spouse/partner relationships, the findings were not statistically significant. This may reflect limitations in sample size, question framing, or analytic sensitivity rather than a true absence of impact. Given the extensive literature documenting marital strain in caregiving contexts, the lack of significant findings here suggests that the study may not have adequately captured the complexity of spousal dynamics. In addition, the study focused primarily on documenting negative impacts without exploring potential coping strategies, resilience factors, or interventions that might mitigate social isolation. Some caregivers may develop adaptive mechanisms or find strength in shared caregiving experiences, but these positive dimensions were not systematically analyzed. As a result, the study presents a predominantly deficit-based perspective.

Tables. No acronyms in title and clarify always space and temporal location of this study, e.g., Children's Hospital Los Angeles and so on ….

The paper has a lot of tables that are difficult to digest, some of them can be put in appendix and inserting in the text the most important ones to improve the readability…

To reiterate, avoiding a lot of sub-headings that create fragmentation of the paper.

Discussion.

First, authors have to synthesize the main results in a simple table to be clear for readers and then show what this study adds compared to other studies. Although the Results section provides a detailed description of the data collected and analyzed, there needs to be a more critical synthesis and comparison of the findings with the literature. The discussion section has to interpret and describe the significance of your findings in relation to what was already known about the research problem being investigated and explain any new understanding or insights that emerged from your research. The discussion has better to connect to the introduction through the research questions and the literature you reviewed. The discussion clearly has to explain how this study advances these research topics and previous study by some of these authors…..Comment on whether the results were expected for each set of findings; go into greater depth to explain unexpected findings. Moreover, either compare your results with the findings from other studies or use the studies to support your results.

A claim for how the results can be applied more generally in other US contexts and also in other nations. Describe lessons learned, proposing recommendations based on incentive systems in Caregiver Social Networks that can help improve the problem and topic under study, highlighting best practices of health and social policies.

Conclusion so short is useless. Conclusion has not to be a summary, but authors have to focus on manifold limitations of this study (moving here the lines 278-293) and not in previous section…. The Conclusion does not adequately discuss the theoretical and health policy implications of the study. Discuss how the gap in the literature has been addressed about the research problem, compared to similar studies.

Make sure you create 3 subsections in the Conclusion: 1) Theoretical Implications, 2) Health Policy Implications based on incentive systems for Caregiver Social Networks, and 3) Future Research.

Overall, then, the paper is interesting, but the structure is confusing. Theoretical framework is weak, and some results create confusion… structure of the paper has to be improved; study design, discussion and presentation of results have to be clarified using suggested comments.

I kindly recommend revising the paper by thoroughly addressing all comments provided, including engaging with the suggested readings. These references should be carefully reviewed and appropriately incorporated into both the main text and the bibliography. I will conduct a detailed evaluation of the revised version, and only then will the paper be considered for further review.

Please note that failure to implement the suggested improvements may result in the paper not being accepted.

Esther Yejin Lee, Nicole Neil, Deanna C. Friesen, 2021. Support needs, coping, and stress among parents and caregivers of people with Down syndrome, Research in Developmental Disabilities,Volume 119,

Sandra Marquis, Michael V. Hayes, Kimberlyn McGrail 2019. Factors Affecting the Health of Caregivers of Children Who Have an Intellectual/Developmental Disability, https://doi.org/10.1111/jppi.12283

Coccia M. 2019. Comparative Incentive Systems. A. Farazmand (ed.), Global Encyclopedia of Public Administration, Public Policy, and Governance, Springer, https://doi.org/10.1007/978-3-319-31816-5_3706-1.

Robert M. Hodapp 2007. Families of persons with Down syndrome: New perspectives, findings, and research and service needs. Mental retardation and developmental disabilities, Volume13, Issue3, Special Issue on Down Syndrome

Roll AE, Bowers BJ. Building and Connecting: Family Strategies for Developing Social Support Networks for Adults With Down Syndrome. Journal of Family Nursing. 2019;25(1):128-151. doi:10.1177/1074840718823578

Coccia M. 2019. Intrinsic and extrinsic incentives to support motivation and performance of public organizations, Journal of Economics Bibliography, vol. 6, no. 1, pp. 20-29, http://dx.doi.org/10.1453/jeb.v6i1.1795

Chung eun Lee, Meghan M. Burke, Catherine K. Arnold, Aleksa Owen, 2018. Comparing differences in support needs as perceived by parents of adult offspring with down syndrome, autism spectrum disorder and cerebral palsy, https://doi.org/10.1111/jar.12521

A. M. Giovannetti, V. Covelli, D. Sattin, M. Leonardi. 2915. Caregivers of patients with disorder of consciousness: burden, quality of life and social support, Acta Neurologica, https://doi.org/10.1111/ane.12392

Reviewer #3: I have to start with thanking the authors for making me aware of the issue at hand, i.e. the (possible) differences between DSRD and 'regular' DS. The authors have made a commendable effort to study the impact for the caregivers, but unfortunately I do not think the manuscript in its current form meets the criteria for publication.

I will be brief in order to bring my major criticism across: the study in this form confuses the assessment of the patients and the viewpoints of the caregivers to a degree that makes the results hard to interpret and understand. This firstly stems from the use of a qualitative measure, and therefore qualitative results that had to be coded into quantitative data in order to be analysed quantitatively. Ideally, this step would have been unnecessary by simply using a quantitative measure. In principal, I think researchers should use quantitative measures if they want to perform quantitative analyses, and use qualitative measures, if they want to establish insight, rather than analysis. But that point has passed and I can therefore look past it.

It is then confusing that the results are split into 'Responses of CDS items' and 'Qualitative responses'. And even that I can look past. However, what I can not look past is that the characteristics of the caregivers, who are the actual research subjects, are not presented or discussed. In order for the reader to understand and interpret the results, they would need to know about the demographics (e.g.: the DSRD patients are significantly older: does that translate into older caregivers?), the family setup (e.g.: how many caregivers are present in the family? How many (older) siblings?) etc. I feel this is a major shortcoming and should be corrected: treat the caregivers as your sample and present their relevant characteristics.

Some additional points:

- a coding process introduces so much fuzziness, that I don't think double decimals are in order

- At the end of the introduction, I don't feel the point that DSRD symptoms are unique compared to symptoms of other neurological diagnoses had been amde sufficiently

- the coding process was not sufficiently described for me. I would like more info about how the number of codes was established and how many disagreements there were between the two coders

- this point may potentially be more important, but I can only interpret that after I know the caregiver characteristics: what would the authors make of the fact that the age difference between DSRD and DSN patients is almost one SD? Could this fact alone explain some of the differences?

**Do you want your identity to be public for this peer review?** For information about this choice, including consent withdrawal, please see our Privacy Policy

Reviewer #1: No

Reviewer #2: No

Reviewer #3: No

---

## [Author Response · Author response to Decision Letter 1]

11 Dec 2025

Reviewer 1:

Comment 1: Participants were recruited from a neurology clinic and a Facebook support group for DSRD caregivers. However, the authors do not specify how many participants came from each group. It can be assumed that the DSRD diagnosis in the neurology clinic group is correct. This is more difficult to assume for the other group, as the survey contains a number of difficult medical terms, and participants were not provided with guidance by one of the researchers when completing it. This could be a potential source of inaccuracy.

> Author Response: Thank you for identifying this clarity issue. The authors have added this information to the results section (first paragraph) as follows:

“In the DSRD cohort, 141 (61.8%) were recruited from clinics and 87 (38.2%) were recruited from online groups compared to 94 (68.6%) and 43 (31.4%) in the DSN cohort, respectively (p=0.19).”

The similarity between the groups and the lack of statistical significance between the percentages recruited from different environments would diminish some of these concerns although it remains possible that difficulty with medical terminology could influence responses.

Comment 2: It is also unclear whether the survey was validated in advance, which could also be a potential source of inaccuracy.

> Author Response: Thank you for identifying this clarity issue. The authors validated the survey for readability, comprehension and time to complete. The authors have added this clarifying statement to the methods section as follows:

“Prior to release of the survey to prospective participants, the survey was validated for readability, comprehension and time to complete by a group of 10 caregivers. Feedback was collected in an open format and modifications for language, user-experience, and organization were made based on the feedback of these caregivers.”

However, we also acknowledge that the survey has not yet undergone full psychometric validation (e.g., factor structure and test–retest reliability in independent samples). We now explicitly note this in the Survey and Qualitative Analysis subsection and reiterate it as a limitation in the Discussion.

The authors have thus added the following statement to the limitations section of the discussion:

“In addition, the tool was partially validated by caregivers for readability and clarity although the authors acknowledge that the survey has not yet undergone full psychometric validation (e.g., factor structure and test–retest reliability in independent samples).”

Comment 3: The authors state that they collected demographic and clinical information via the survey. However, these questions are not included in Appendix A.

> Author Response: Thank you for this comment. The authors have amended the Appendix to include this information with the full survey. This appears to have been a Redcap formatting issue for which the authors apologize.

Comment 4: Table 2 lists the mean age, gender, ethnicity, race, and mean duration of symptoms. However, in the Results section, no distinction is made according to ethnicity or race regarding the impact on caregivers. Thus, the significance of mentioning this in Table 2 is unclear.

> Author Response: Thank you for this comment. The authors note that as there was no statistical significance between the groups with regards to race (p=0.78) and ethnicity (p=0.37). The data reported in the results section was already adjusted for race, ethnicity, and age. This has now been explicitly stated in the methods section as follows:

“A p-value of less than 0.05 was considered to be statistically significant, with values trending toward this threshold noted for further consideration. All p-values are reflected of adjusted analysis to control for age, ethnicity and race.”

The authors apologize for this confusion.

Comment 5: I also miss the mention of how many of the DSRD group have already been treated and what the outcome was. It would have been interesting to analyze whether that would make a difference to the QOL of the caregivers.

> Author Response: Thank you for this comment. This information was unfortunately not collected in the original survey. As it can take several years to acquire clinical stability and improvements and there is wide heterogeneity in treatments offered regionally, this response could have been heavily skewed and may lead to over-interpretation of “treatment” in this context.

The authors agree that this is a logical next step as it would be presumed that quality of life would improve over time.

Given the impact of this, the authors have mentioned this explicitly in the limitations section of the discussion as follows:

“In addition, treatment data was not collected due to the lack of ability to verify interventions, timing of interventions, and the lack of ability to adjust for regional therapeutic offering differences. The authors believe that successful treatment of DSRD would improve QOL and this represents a future area of study for the team.”

Comment 6: It is also never mentioned who the caregiver is who completes the survey.

> Author Response: Thank you for identifying this clarity issue. The authors have added this information to the results section as follows:

“Those completing the survey were most often mothers (166 (72.8%) in DSRD v. 106 (77.4%) in DSN), fathers (43 (18.9%) in DSRD v. 22 (16.1%) in DSN), or guardians (11 (4.8%) in DSRD v. 9 (6.7%) in DSN) (p=0.61).”

Comment 7: I have another question about Table 2. The mean duration of symptoms in the DSN group is 4.59 years. This surprises me because the authors describe these individuals as young people with ASD, epilepsy, or moyamoya vasculopathy.

> Author Response: Thank you for this comment. The authors note that the diagnosis of autism spectrum disorder in individuals with Down syndrome is heterogeneous and ranges from 3-8 years of age. Both epilepsy and moyamoya vasculopathy are bimodal in their distribution and can happen early or later in childhood. The duration of symptoms for the DSN group corresponds to the years since diagnosis of this condition, not the age at which it was diagnosed. Table 2 now has additional information regarding this in the legend to improve clarity as follows:

“α: indicates years since the diagnosis of this condition”

It is possible that individuals who have an onset of disease at a younger age could be more or less difficult to handle as a caregiver. This is a possible interpretation of the data as well although there is no literature to support this is an issue in individuals with Down syndrome. Unfortunately, there are no other perfectly matched conditions affecting individuals with DS in the late teen years for which to compare DSRD.

The authors have performed a separate analysis to confirm this hypothesis which is not listed in the results section as follows:

“Caregiver distress did not worsen with increasing age in either cohort. In the DSRD group, age was not associated with higher odds of endorsing worsening caregiver outcomes (per 1-year increase: OR 1.01; 95% CI 0.98–1.04; p = 0.58). Similarly, in the DSN group, age was not associated with worsening caregiver outcomes (per 1-year increase: OR 0.99; 95% CI 0.95–1.03; p = 0.66). Results were unchanged after adjustment for sex and symptom duration (DSRD adjusted OR 1.00, 95% CI 0.97–1.03, p = 0.91; DSN adjusted OR 0.98, 95% CI 0.94–1.02, p = 0.31).”

Comment 8: The text does not specify how many people in the DSN group have each diagnosis.

> Author Response: Thank you for identifying this clarity issue. The authors have added this information to Table 2.

Comment 9: Although the DSN group is younger on average (mean age of 16.05 years), I find it strange that the mean symptom duration is only 4.59 years. A diagnosis of ASD and epilepsy in children is often made at a younger age. Perhaps the authors should verify this. I also wonder how meaningful the comparison between the DSRD and DSN groups is.

> Author Response: Thank you for this comment. As noted above, the reviewer is correct in identifying this theme. The authors note that the diagnosis of autism spectrum disorder in individuals with Down syndrome is heterogeneous and ranges from 3-8 years of age. Both epilepsy and moyamoya vasculopathy are bimodal in their distribution and can happen early or later in childhood. In addition, atlanto-axial instability is often diagnosed in the teen years. These data points are subjectively reported by family members which could introduce a recall bias, as noted in the discussion’s limitations section.

A challenge with this cohort is that responses were anonymous to the research team, making verification of diagnoses not possible. This has been made explicit in the limitations section of the discussion:

“Additionally, the DSN comparison group was heterogeneous, encompassing various neurologic conditions, which may obscure disorder-specific effects and limit direct comparisons. This is further complicated by this study’s lack of ability to clinically confirm diagnoses.”.

As noted in the methods, this was specifically detailed to those completing the survey although was not verifiable:

“In the DSN group, the diagnosis of the neurological condition was based solely on caregiver self-report, and diagnosis by a physician was required in all cases.”

The authors agree that the meaningfulness of comparison of DSRD to DSN is not perfect but represented a reasonable comparator of medical complexity in the same underlying disease state. It allows interpretation to not simply be related to “just Down syndrome” effect.

As noted above, the authors have added an additional analysis to demonstrate that age did not influence caregiver responses:

“Caregiver distress did not worsen with increasing age in either cohort. In the DSRD group, age was not associated with higher odds of endorsing worsening caregiver outcomes (per 1-year increase: OR 1.01; 95% CI 0.98–1.04; p = 0.58). Similarly, in the DSN group, age was not associated with worsening caregiver outcomes (per 1-year increase: OR 0.99; 95% CI 0.95–1.03; p = 0.66). Results were unchanged after adjustment for sex and symptom duration (DSRD adjusted OR 1.00, 95% CI 0.97–1.03, p = 0.91; DSN adjusted OR 0.98, 95% CI 0.94–1.02, p = 0.31).”

Comment 10: The social media quotes are both instructive and moving.

> Author Response: Thank you for this comment. While not typical to include for a study like this the authors agreed that this was important to highlight.

Comment 11: Despite its methodological limitations, the article demonstrates that caring for someone with DSDR significantly impacts the social lives of caregivers. However, I have difficulty with the word "permanent" when it comes to social ties. I am not a native English speaker, so perhaps I have misunderstood the meaning of "permanent." To me, it means "lasting forever," which this cross-sectional study with imperfect methodology cannot demonstrate. Perhaps the authors mean something else; I would like to know what that is.

> Author Response: Thank you for this comment. The authors agree that the language of “permanent” is a bit challenging to interpret. The authors agree that this language can be confusing but is designed to reflect a permanence related to the condition. For instance, if there where no changes to a patients condition, would the social network improve. If the answer to this question would be “no” then this would be defined as permanent in this study.

The authors attempted to try different “less permanent” language in the validation trial but caregivers who were evaluating the survey advised that this would be the most easily interpreted for those completing the survey.

The authors appreciate this feedback and will workshop this for future survey-based studies. Taking a more international English language approach may be beneficial for future studies given the reach of DSRD. The following has been added to the limitations section to help clarify this to some extent:

“In addition, while the thematic coding process was consensus-based and methodologically rigorous, qualitative analysis remains inherently interpretive and may not fully capture the complexity of caregiver experiences. Further, the narrower nature of the coding may have oversimplified themes or omitted other areas of social support or strain. In addition, some language such as the use of the term “permanent” or “loss of community participation” can be hard to interpret both for those completing the survey and the researchers coding the data.”

Comment 12 (Minor Remark): It appears that the text on lines 116-117 is repeated on lines 118-119.

> Author Response: Thank you for this comment. The authors have removed duplicate lines in the text.

Comment 13 (Minor Remark): Line 179 states: "The DSRD and DSN objective responses are summarized in Table 4." Does "objective responses" mean the same as "descriptive quotations," as written on line 183? If so, it would be better to replace "objective responses" with "descriptive quotations" in the sentence.

> Author Response: Thank you for identifying this clarity issue. The authors have amended this to be consistent and this line now reads “qualitative responses” as follows:

“The DSRD and DSN qualitative responses are summarized in Table 3”

The authors have also amended the quotations section at the end of the results section to read as follows:

“A table of representative descriptive quotations illustrating these themes and providing a deeper qualitative perspective is included in Table 4.”

Comment 14 (Minor Remark): The figures are difficult to read.

> Author Response: The authors apologize for this issue. We have increased the dpi to increase clarity and readability.

------------------

Reviewer 2:

Comment 1: Title, clarify that is a case study on Children's Hospital Los Angeles and period under study.

> Author Response: We appreciate this suggestion. To provide clearer context while keeping the title concise, we have revised it to:

“Impact of Down Syndrome Regression Disorder on Caregiver Social Networks: A Cross-sectional Study at a Tertiary Pediatric Center in the United States.”

The Abstract and Methods now explicitly state that participants were recruited from Children’s Hospital Los Angeles and a DSRD Facebook support group between July 6 and July 28, 2024 as follows:

“This study had two separate groups of participants: 1) DSRD Cohort: This group consisted of caregivers of individuals with a confirmed or probable diagnosis of DSRD. They were recruited through a neurology clinic at Children's Hospital Los Angeles and a private Facebook support group for DSRD caregivers. The recruitment period lasted 3 weeks (July 6, 2024, to July 28, 2024). 2) DSN Cohort: This group was composed of caregivers of individuals with Down syndrome who also had other active neurological conditions, such as moyamoya vasculopathy, epilepsy, or autism spectrum disorder.”

Comment 2: Abstract has to clarify the goal, and specific health and social implications to face DSRD.

> Author Response: Thank you for identifying this clarity issue. The authors have amended the last sentence of the background section to better clarify this. This now reads as follows:

“The goal of this study was to evaluate the impact of DSRD on caregiver social networks by comparing their experiences to those of caregivers of individuals with Down syndrome and other neurological disorders (DSN).”

Comment 3: Introduction has to better clarify the research questions of this study, indicating the gap presents in literature that this study endeavors to cover, and provide more theoretical background. Clarify differences with this study cited (Caregiver burden and familial impact in Down Syndrome Regression Disorder. Orphanet J Rare Dis 20, 126).

After that authors can focus on the topics of this study to provide a correct analysis for fruitful discussion. (Please refer to the suggested readings, all of which should be thoroughly reviewed and incorporated into your text.).

> Author Response: Thank you for identifying this clarity issue. The authors now have revised the introduction to clearly describe exis

---

## [Decision Letter · Decision Letter 1]

6 Jan 2026

Dear Dr. Santoro,

However, before the manuscript can be considered for acceptance, further revision is required to ensure that the interpretation and framing of the findings are fully aligned with the nature of the data collected.

In particular, the current manuscript frequently presents subjective, cross-sectional, self-reported perceptions as if they reflect objective and structural properties of caregiver social networks. The study does not directly measure social network structure, size, density, or dynamics, but rather caregiver perceptions and narratives regarding social experiences. The manuscript should therefore consistently frame findings as perceived or self-reported social impacts, and avoid language implying objective, causal, or structural network change.

Related to this, the repeated use of the term “permanent” to describe social loss is not supported by the cross-sectional design and should be revised or softened. Claims regarding permanence, progression, or enduring network contraction must be clearly presented as caregiver perceptions rather than empirically demonstrated longitudinal outcomes.

Finally, interpretations comparing the DSRD and DSN groups should be further tempered to reflect the heterogeneity and self-reported nature of the DSN comparator, and the limitations of causal or disorder-specific inference.

Addressing these conceptual and interpretive issues through careful revision of language, framing, and conclusions will substantially strengthen the scientific accuracy and credibility of the manuscript.

plosone@plos.org . A letter that responds to each point raised by the academic editor and reviewer(s). You should upload this letter as a separate file labeled 'Response to Reviewers'.A marked-up copy of your manuscript that highlights changes made to the original version. You should upload this as a separate file labeled 'Revised Manuscript with Track Changes'.An unmarked version of your revised paper without tracked changes. You should upload this as a separate file labeled 'Manuscript'.

We look forward to receiving your revised manuscript.

Kind regards,

Inge Roggen, M.D., Ph.D.

Academic Editor

PLOS One

**Journal Requirements:**

Reviewers' comments:

Reviewer's Responses to Questions

**Comments to the Author**

Reviewer #1: All comments have been addressed

Reviewer #2: (No Response)

2. Is the manuscript technically sound, and do the data support the conclusions?

Reviewer #1: Yes

Reviewer #2: No

3. Has the statistical analysis been performed appropriately and rigorously?

Reviewer #1: I Don't Know

Reviewer #2: No

4. Have the authors made all data underlying the findings in their manuscript fully available?

Reviewer #1: Yes

Reviewer #2: No

5. Is the manuscript presented in an intelligible fashion and written in standard English?

Reviewer #1: Yes

Reviewer #2: Yes

Reviewer #1: (No Response)

Reviewer #2: Dear Authors, although the revision, from this paper is difficult to grasp the scientific advances.

Hence, current manuscript fails to clearly demonstrate any significant scientific advancement

**Do you want your identity to be public for this peer review?** For information about this choice, including consent withdrawal, please see our Privacy Policy

Reviewer #1: No

Reviewer #2: No

---

## [Author Response · Author response to Decision Letter 2]

8 Jan 2026

January 9th, 2026

Dear Editors of PLoS One and Dr. Roggen,

Thank you for the opportunity to revise our manuscript entitled “Impact of Down Syndrome Regression Disorder on Caregiver Social Networks: A Cross-Sectional Study at a Tertiary Pediatric Center in the United States”. The authors sincerely appreciate the time spent in review of this manuscript and have responded in a point-by-point manner to the feedback provided below. The authors believe that the feedback provided, particularly by the editor, significantly improve the quality of the manuscript and we hope that the revised version is acceptable for publication.

Most Sincerely,

Jonathan D. Santoro, MD

--------

Editor: The current manuscript frequently presents subjective, cross-sectional, self-reported perceptions as if they reflect objective and structural properties of caregiver social networks. The study does not directly measure social network structure, size, density, or dynamics, but rather caregiver perceptions and narratives regarding social experiences.

Editor Comment 1: The manuscript should therefore consistently frame findings as perceived or self-reported social impacts, and avoid language implying objective, causal, or structural network change. Related to this, the repeated use of the term “permanent” to describe social loss is not supported by the cross-sectional design and should be revised or softened. Claims regarding permanence, progression, or enduring network contraction must be clearly presented as caregiver perceptions rather than empirically demonstrated longitudinal outcomes.

Authors Response: Thank you for this comment. The authors sincerely appreciate this feedback and agree with this need for more clear distinction. The authors have made the following changes to the manuscript to address the editors comments:

1. The title of the manuscript has been changed to highlight that the data presented is self-reported, not objectively obtained. The title now reads “Caregiver-Reported Social Impacts in Down Syndrome Regression Disorder”

2. In the abstract, the authors have clarified that prior studies have also focused on subjective report of negative impacts. The 2nd sentence of the background section of the abstract now reads as follows: “DSRD increases demands on caregivers, leading to sleep disturbances, financial distress, and negative impacts on caregiver-reported social connections and perceived social support. The goal of this study was to characterize the caregiver-reported impacts of DSRD on social networks by comparing their experiences to those of caregivers of individuals with DS and other neurological disorders (DSN).”

3. In the methods section of the abstract, the authors have also specified the self-perception component of the fields studied. This section now reads as follows: “Participants completed the DSRD Caregiver Distress Survey (CDS), which included four qualitative, open-ended questions focused on self-perception of adult friendships, social network impact, spouse/partner impact, and social network size.”

Further, this was also specifically stated in the abstract methods as follows: “Responses were analyzed using thematic coding; resulting theme frequencies summarize caregiver-reported perceptions and narratives and do not represent objectively measured social network structure.”

4. In the abstract, the permanent statement in the results has been replaced and softened as requested. This sentence now reads as follows: “Caregivers in the DSRD group were significantly more likely to report “Social Withdrawal and Isolation” (43.2% vs. 17.9%, p=0.006), “Loss of Community Participation and/or Support” (16.7% vs 4.5%, p=0.043) and a "Perceived Enduring Loss of Social Ties" (35.3% vs. 8.7%, p=0.002) compared to the DSN group.”

Of note, the authors have also used this terminology throughout the manuscript, figures, and tables in multiple locations.

5. In the abstract’s conclusion, the verbiage of permanence and marked impact has also been softened. The first sentence now reads as follows: “This study's findings reveal a significant and complex process of perceived social disengagement among caregivers describing social withdrawal and loss of social ties that they experienced as enduring.”

6. In the introduction, the authors have made clearer that the purpose of this study was characterization of self-reported social impacts. This sentence now reads as follows:

“The purpose of this study was to characterize caregiver-reported social impacts associated with caring for an individual with DSRD and compare these reported impacts with a heterogeneous caregiver group of individuals with DS and other neurological disorders (DSN) [12]. This study also characterized caregiver perceptions regarding changes in the quality of adult friendships, impact on social networks, impact on spouse/partner, and social network size.

This has also been clarified in the introduction further as follows: “Understanding how the unique demands of DSRD caregiving may be associated with caregiver perceptions of reduced social support, is important in highlighting the overall care burden and mental health outcomes.”

7. To allow for persistent use of caregiver-reported results, the authors have also made this edit in the results section: “Caregiver-reported results indicated a significantly higher percentage of DSRD caregivers reported a 'Perceived Enduring Loss of Social Ties' (35.3% vs. 8.7%, p=0.01, [0.1, 0.4]) compared to the DSN group (Figure 2b).”

8. The authors have also revised the term “objective responses” in reference to Table 4 as follows in the results section: “The DSRD and DSN self-reported responses are summarized in Table 4.”

9. In the first paragraph of the discussion, the introductory sentence has been tempered to read as follows: “In this cross-sectional comparison of caregivers of individuals with DSRD versus DSN, we found consistent caregiver-reported perceptions of social strain and reduced social connectedness across multiple domains, with the clearest between-group differences involving social withdrawal, reduced community participation, and perceived enduring loss of ties.”

10. In the discussion, another highlighting of self-report was made with regards to network size. This sentence now reads as follows: “This distinction is crucial because although caregivers across many conditions in the DSN group face obstacles to community participation, DSRD caregivers more frequently reported social losses that they experienced as enduring; however, persistence and temporality cannot be determined from this cross-sectional survey. This degree of perceived enduring social loss may reflect higher caregiving burden among DSRD caregivers in this respondent sample.”

11. In the third sentence, the authors have also clarified the self-perception aspect of this and added in increased distinction between the DSRD and DSN cohorts (point two by the editor): “Prior caregiver studies in neurodevelopmental and neurologic conditions describe isolation, stigma, and reduced participation; our contribution is describing caregiver-reported social impacts in the DSRD cohort relative to a heterogeneous DSN comparison group. While between-group differences were observed in reported community participation and reported loss of social ties, these findings should be interpreted cautiously given the self-reported nature of outcomes and the heterogeneity of neurologic diagnoses within the DSN comparator, which limits disorder-specific inference [12]. While these differences are statistically significant within this sample, they represent subjective narratives and should not be used to infer structural network differences between disorders.”

12. Further, the authors have softened the findings reported in the 2nd paragraph of the discussion and added that losses were not permanent but perceived as enduring as follows: “The most statistically significant findings in this study centered on caregivers’ descriptions of loss of social ties that they experienced as enduring.”

Further in this paragraph: “This diminished capacity for social engagement is not merely a temporary inconvenience but was described by caregivers as contributing to an enduring loss of social ties.”

and, removed the concept of permanence from this sentence:

“More broadly, these results suggest that the challenges associated with caregiving lead to significant consequences that may alter the caregiver’s social experiences and perceived support.”

and

“This distinction is crucial because although caregivers across many conditions in the DSN group face obstacles to community participation, DSRD caregivers more frequently reported social losses that they experienced as enduring; however, persistence and temporality cannot be determined from this cross-sectional survey. This degree of perceived enduring social loss may reflect higher caregiving burden among DSRD caregivers in this respondent sample.”

and

“We cannot attribute perceived caregiver social changes to any single factor in this cross-sectional design as prior work on social networks being “static” in people with DS may not generalize to caregiver networks [23]”

13. In the limitations section, the authors have added an explicit statement about lack of social network size measurements. This statement reads as follows: “Importantly, we did not directly measure social network structure (size, density, or dynamics) using formal social network analysis; findings therefore reflect caregiver perceptions and narratives rather than objectively observed network properties or causal change.”

14. In the conclusions, the authors have made more straightforward statements regarding the reductions as follows: “Across domains (friendships, partner relationships, network size), caregivers described perceived reductions in social connectedness and participation.”

and

“Longitudinal study design will also afford the ability to track impacts over treatment periods which may yield unique fluctuations and assess if self-reported perceptions of loss are as enduring as reported in this study.”

Editor Comment 2: Finally, interpretations comparing the DSRD and DSN groups should be further tempered to reflect the heterogeneity and self-reported nature of the DSN comparator, and the limitations of causal or disorder-specific inference.

> Authors Response: Thank you for this comment. The discussion has been edited in a number of locations to provide more softened interpretations and highlight the limitations of causal and/or disorder specific inference. The following edits have been made to the discussion:

“Prior caregiver studies in neurodevelopmental and neurologic conditions describe isolation, stigma, and reduced participation; our contribution is describing caregiver-reported social impacts in the DSRD cohort relative to a heterogeneous DSN comparison group. While between-group differences were observed in reported community participation and reported loss of social ties, these findings should be interpreted cautiously given the self-reported nature of outcomes and the heterogeneity of neurologic diagnoses within the DSN comparator, which limits disorder-specific inference [12].”

and, to lead off the 3rd paragraph:

“Comparisons between caregivers of individuals with DSRD and those in the DSN group should be interpreted cautiously.”

and in the same paragraph:

“While similar themes of social isolation, decreased community participation, and relationship strain are reported by caregivers of children with other neurological disorders, in this respondent sample, caregivers in the DSRD group more frequently perceived constraints on social engagement than caregivers in the DSN group.”

In the limitations sections, the authors have also expanded our first point to read as follows: “First, reliance on caregiver self-report introduces the possibility of recall bias and social desirability bias and limits diagnostic verification, particularly in the DSN group. Because the DSN comparator is heterogeneous and diagnoses/outcomes were not clinically adjudicated within this study, between-group differences should be interpreted as differences in caregiver-reported experiences among respondents rather than disorder-specific effects attributable uniquely to DSRD.”

This has also been expanded further down to state: “Additionally, the DSN comparison group was intentionally heterogeneous, encompassing various neurologic conditions; this limits direct comparability and constrains disorder-specific inference, as observed differences may reflect differences in diagnosis mix, comorbidity burden, severity, and caregiver context rather than DSRD itself.”

and

“Additionally, the cross-sectional design and lack of longitudinal follow-up preclude assessment of causality or temporal dynamics; accordingly, findings should be interpreted as associative caregiver reports rather than evidence of causal or progressive disorder-specific social network change.”

Finally, the authors have changed the framework of the conclusions section (theoretical implications) as follows: “By comparing caregiver reports in the DSRD cohort with those from a heterogeneous DSN group using a shared framework, we provide initial descriptive evidence of differential perceived social impacts; however, disorder-specific inference will require longitudinal studies with more diagnostically homogeneous comparators and clinically verified diagnoses.”

-----

Reviewer 1: All comments were addressed

----

Reviewer 2: (no response)

---

## [Editor Report · Decision Letter 2]

13 Jan 2026

Dear Dr. Santoro,

Thank you for submitting your manuscript to PLOS ONE. After careful consideration, we feel that it has merit but does not fully meet PLOS ONE’s publication criteria as it currently stands. Therefore, we invite you to submit a revised version of the manuscript that addresses the points raised during the review process.

There remain some smaller issues that are not fully addressed:

- remove “ties”

- remove “network size”, replace with “relationships,” “contacts,” “perceived shrinkage of social world”

- rename the outcome construct entirely

 and explicitly state: This is a narrative burden-of-care study, not a network study.

We look forward to receiving your revised manuscript.

Kind regards,

Inge Roggen, M.D., Ph.D.

Academic Editor

PLOS One
---

## [Author Response · Author response to Decision Letter 3]

15 Jan 2026

January 15th, 2026

Dear Editors of PLoS One and Dr. Roggen,

Thank you for the opportunity to revise our manuscript entitled “Caregiver-Reported Social Impacts in Down Syndrome Regression Disorder”. The authors sincerely appreciate the time spent in review of this manuscript and have responded in a point-by-point manner to the feedback provided below. The authors believe that the feedback provided, particularly by the editor, significantly improve the quality of the manuscript and we hope that the revised version is acceptable for publication.

Most Sincerely,

Jonathan D. Santoro, MD

------

Editor’s Comments: There remain some smaller issues that are not fully addressed:

Editor Comment 1: Remove “ties”

> Author's Response: Thank you for this comment. The authors have removed the term “ties” and replaced it with “connections” in multiple instances. This has also been reflected in associated tables and figures.

Editor Comment 2: Remove “network size”, replace with “relationships,” “contacts,” “perceived shrinkage of social world”

> Author's Response:

Thank you for this comment. The authors have made the requested edits with the following changes reflected in the text:

Abstract:

“The goal of this study was to characterize the caregiver-reported impacts of DSRD on social relationships by comparing their experiences to those of caregivers of individuals with DS and other neurological disorders (DSN).”

“Participants completed the DSRD Caregiver Distress Survey (CDS), which included four qualitative, open-ended questions focused on self-perception of adult friendships, social relationship impact, spouse/partner impact, and perceived social contacts size.”

“In the DSRD cohort, a high-level overview revealed that 65.66% of responses reported a negative impact on adult friendships, while 71.21% reported a negative impact on social relationships.”

“A negative impact on spouse/partner relationships was reported in 51.53% of responses, and a perceived shrinkage of social world was found in 52.82%.”

Introduction:

“This can cause caregivers to experience sleep disturbances, financial distress, and negative impacts to family dynamics and social relationships, among other challenges. The time-intensive nature of care demands means that individuals with DSRD require significantly more support than they did prior to their onset of symptoms, with potential cascading negative effects on caregivers and their social relationships. While previously published literature has identified caregiver burden, dedicated exploration of the impact on social relationships of caregivers is unknown.”

“Additionally, caregivers have reported increased social exclusion and stigma, resulting in a significant decrease of their social relationships by both contacts and quality”

“This study also characterized caregiver perceptions regarding changes in the quality of adult friendships, impact on social contacts, impact on spouse/partner, and social relationships”

Methods:

“The survey included a section called the DSRD Caregiver Distress Survey (CDS) which featured four specific open-ended qualitative questions designed to capture the nuanced impact of caregiving on a caregiver's social relationships”

“We analyzed caregiver free-text responses to four open-ended survey items addressing impacts of the individual’s condition on adult friendships, marital/partner relationships, perceived social support, and perceived changes in social relationships.”

“We conducted a semantic thematic analysis using a hybrid inductive–deductive coding strategy: an initial deductive code “start list” was derived from the survey domains (e.g., friendship disruption, partner strain, social support changes, social relationship and contact contraction), and inductive codes were added iteratively to capture unanticipated content (e.g., stigma-related withdrawal, time burden, role overload, health system navigation).”

Results:

“Notably, caregivers of individuals with DSRD were more likely to report perceived declines in the quality of adult friendships (64.4% vs. 29.2%, p < 0.001), overall social relationships (71.1% vs. 34.3%, p < 0.001), and marital relationships (52.0% vs. 19.7%, p < 0.001) compared with DSN caregivers.”

“Additionally, a greater proportion of DSRD caregivers reported a perceived shrinkage of their social world (54.1% vs. 38.0%, p = 0.01).”

“Out of 198 total responses regarding the Quality of Adult Friendships, 130 (65.6%) reported a negative impact. For Social Relationships Impacted, 141 out of 198 responses (71.2%) indicated a negative impact. Regarding Spouse/Partner Impact, 101 out of 196 responses (51.5%) were negatively impacted. Finally, for Perceived Shrinkage of Social World, 103 of 195 responses (52.8%) reported a negative impact. The DSRD and DSN self-reported responses are summarized in Table 4.”

“Qualitative responses regarding social relationships were provided by 100 DSRD participants and 44 DSN participants. Caregivers reported impacts on various aspects of their social interactions (Figure 2b).”

“Regarding perceived social contacts, 51 DSRD caregivers and 46 DSN caregivers provided written responses.”

“"Reduced Invitations from Social Contacts" was more prevalent in the DSRD group (19.6% vs. 8.7%, p=0.38, [-0.1, 0.2]).”

Discussion:

“Caregivers described caregiving demands as associated with reduced social engagement and fewer opportunities to sustain friendships and community participation, which, when repeatedly reinforced, were perceived as progressive contact/relationship loss.”

“These challenges align with a broader finding that caregiving constraints, particularly a lack of time and energy, affect a caregiver’s ability to maintain their social relationships.”

“Likewise, research on caregivers in the broader DSN population, such as mothers of children with various long-term care needs, highlights similar challenges including social isolation and distance from friends stemming from a lack of knowledgeable support systems and gaps in respite services”

“Caregivers more frequently reported a perceived shrinkage of the social world in the DSRD cohort, primarily driven by a time and energy constraints and reduced invitations from existing social relationships.”

“It is the author’s hypothesis that this pattern is a negative feedback loop wherein higher levels of care for loved ones with DSRD leads to decreased time and ability to engage socially which subsequently leads to decreased invitations for engagement from the existing social relationships.”

“We cannot attribute perceived caregiver social changes to any single factor in this cross-sectional design as prior work on social relationships being “static” in people with DS may not generalize to caregiver relationships [23].”

“In addition, this study is limited by its cross-sectional design as social impacts (and caregiver burden) may change over time.”

“The authors feel strongly that certain factors such as socioeconomic status may be important drivers of perceived shrinkage of the social world and warrant exploration in the future as discrepancies with primary caregivers not having financial resources to obtain additional care support.”

“Additionally, the cross-sectional design and lack of longitudinal follow-up preclude assessment of causality or temporal dynamics; accordingly, findings should be interpreted as associative caregiver reports rather than evidence of causal or progressive disorder-specific social relationship change.”

Conclusions:

“This study advances the DSRD caregiving literature by focusing on caregiver social relationships (not only global burden). Across domains (friendships, partner relationships, social contacts), caregivers described perceived reductions in social connectedness and participation”

Table 1:

1. Column 2 title change to: Impact on Social Relationships

2. Decreased Social Contacts

3. Column 4 title change to: Perceived Shrinkage of Social World

4. Reduced Invitations from Social Contacts

5. Shift to Condition-Specific Support Relationships

Table 3:

1. Subcategory title changed to: Social contacts and relationships.

2. Social relationships impacted

3. Perceived shrinkage of social world. The authors also changed this to a yes/no response to match the verbiage change.

Table 5:

1. Title changed to: Qualitative findings on the social relationships of DSRD caregivers.

2. Impact on social networks changed to: Impact on Social Relationships.

3. Social network size changed to: Perceived Shrinkage of Social World.

Editor Comment 3: Rename the outcome construct entirely

> Author's Response: Thank you for this comment. As noted above, the verbiage changes have necessitated changes to the outcome constructs which are now reflected throughout the manuscript and are highlighted in the section above.

Specifically, the four main fields have been renamed: 1) Quality of Adult Friendships, 2) Impact on Social Relationships, 3) Spouse/Partner Impact, and 4) Perceived Shrinkage of Social World. The sub-sections of these have also been modified in the Impact on Social Relationships (Decreased Social Contacts), and Perceived Shrinkage of Social World (Reduced Invitations from Social Contacts, Shift to Condition-Specific Support Relationships, and Perceived Enduring Loss of Social Contacts).

This is now clearly reflected in Table 1. In addition, Figures 1, 2, and 3, and their associated legends, have been updated to reflect this as well.

Editor Comment 4: Explicitly state: This is a narrative burden-of-care study, not a network study.

> Author's Response: Thank you for this comment. The authors have made the suggested edits, verbatim, as follows:

Abstract:

“This is a narrative burden-of-care study, not a network study. Using cross-sectional study design, caregivers of individuals with DSRD (n=228) and DSN (n=137) were recruited from a neurology clinic and a DSRD Facebook support group.”

Methods:

“This is a narrative burden-of-care study, not a network study and utilized a cross-sectional study design.”

---

## [Editor Report · Decision Letter 3]

19 Jan 2026

Caregiver-Reported Social Impacts in Down Syndrome Regression Disorder

PONE-D-25-57249R3

Dear Dr. Santoro,

We’re pleased to inform you that your manuscript has been judged scientifically suitable for publication and will be formally accepted for publication once it meets all outstanding technical requirements.

Kind regards,

Inge Roggen, M.D., Ph.D.

Academic Editor

PLOS One
---

## [Editor Report · Acceptance letter]

PONE-D-25-57249R3

PLOS One

Dear Dr. Santoro,

I'm pleased to inform you that your manuscript has been deemed suitable for publication in PLOS One. Congratulations! Your manuscript is now being handed over to our production team.

Kind regards,

on behalf of

Prof. Inge Roggen

Academic Editor

PLOS One